



# On the visual detection of non-natural records in streamflow time series: challenges and impacts

Laurent Strohmenger[1], Eric Sauquet[2], Claire Bernard[3], Jérémie Bonneau[2], Flora Branger[2], Amélie Bresson[4], Pierre Brigode[1,5], Rémy Buzier[6], Alban de Lavenne[1], Olivier Delaigue[1], Alexandre Devers[2], Guillaume Evin[7], Maïté Fournier[8], Shu-Chen Hsu[1], Sandra Lanini[9,10], Thibault Lemaitre-Basset[1,11], Claire Magand[12], Guilherme Mendoza Guimarães[1], Max Mentha[13], Simon Munier[14], Charles Perrin[1], Tristan Podechard[15], Léo Rouchy[2], Malak Sadki[14], Myriam Soutif-Bellenger[1,16], François Tilmant[1], Yves Tramblay[17], Anne-Lise Véron[1], Jean-Philippe Vidal[2], and Guillaume Thirel[1]

[1]Université Paris-Saclay, INRAE, HYCAR Research Unit, Antony, France
[2]INRAE, UR RiverLy, Villeurbanne, France
[3]Chambre d'agriculture du Vaucluse, Avignon, France
[4]EPIDOR, Castelnaud-la-Chapelle, France
[5]Université Côte d'Azur, Observatoire de la Côte d'Azur, CNRS, OCA, IRD, Géoazur, Sophia-Antipolis, France
[6]University of Limoges, URA IRSTEA, Limoges, France
[7]Univ. Grenoble Alpes, INRAE, CNRS, IRD, Grenoble INP, IGE, Grenoble, France
[8]ACTeon – Environment, Research & Consultancy, Grenoble, France
[9]BRGM, unité EAU-RMD, Montpellier, France
[10]G-EAU, UMR 183, INRAE, CIRAD, IRD, AgroParisTech, Supagro, BRGM, Montpellier, France
[11]UMR 7619 METIS, Sorbonne Université, CNRS, EPHE, Paris, France
[12]Office français de la biodiversité (OFB), Vincennes, France
[13]Safege-Suez Consulting, Paris, France
[14]CNRM, Université de Toulouse, Météo-France, CNRS, Toulouse, France
[15]CEREG, Montpellier, France
[16]AgroParisTech, 75005, Paris, France
[17]HSM, University of Montpellier, CNRS, IRD, IMT, Montpellier, France

**Correspondence:** Laurent Strohmenger (Laurent.strohmenger@inrae.fr) and Guillaume Thirel (Guillaume.thirel@inrae.fr)

**Abstract.** Large datasets of long-term streamflow measurements are widely used to infer and model hydrological processes. However, streamflow measurements may suffer from what users can consider as anomalies, i.e., non-natural records that may be erroneous streamflow values or anthropogenic influences that can lead to misinterpretation of actual hydrological processes. Since identifying anomalies is time consuming for humans, no study has investigated their proportion, temporal distribution, and influence on hydrological indicators over large datasets. This study summarizes the results of a large visual inspection campaign of 674 streamflow time series in France made by 43 evaluators, who were asked to identify anomalies falling under five categories, namely, linear interpolation, drops, noise, point anomaly, and other. We examined the evaluators' individual behavior in terms of severity and agreement with other evaluators, as well as the temporal distributions of the anomalies and their influence on commonly used hydrological indicators. We found that inter-evaluator agreement was surprisingly low, with an average of 12 % of overlapping periods reported as anomalies. These anomalies were mostly identified as linear interpolation





and noise, and they were more frequently reported during the low-flow periods in summer. The impact of cleaning data from the identified anomaly values was higher on low-flow indicators than on high-flow indicators, with change rates lower than 5 % most of the time.

We conclude that the identification of anomalies in streamflow time series is highly dependent on the aims and skills of each
evaluator, which raises questions about the best practices to adopt for data cleaning.

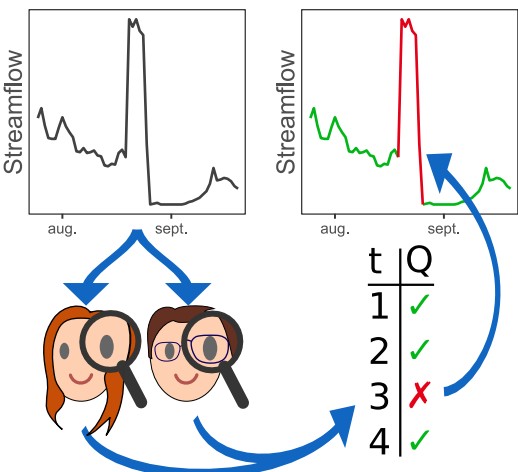

# 1 Introduction

Water is essential for the well-being of our societies, as it supports recreational activities, biodiversity, agriculture, industrial development, and fresh water supply. Yet water, and also lack of water, can become a threat during extreme events such as
floods (Merz et al., 2021) and droughts (Blauhut et al., 2022). This highlights the importance of management strategies based on scientific understanding of hydrological processes in order to mitigate their impacts. The starting point of the learning framework proposed by Dunn et al. (2008) is the acquisition of field data (e.g., river streamflow) to hypothesize and conceptualize the functioning of a catchment before making predictions.

Acquiring long-term data is a crucial step in studying the water cycle and its interactions with natural (i.e., without human
influences) and anthropogenic drivers in the atmosphere, biosphere, and lithosphere (Gaillardet et al., 2018). A well-distributed monitoring network is required to cover the spatial heterogeneity of these interactions within large territories and to improve the robustness of statistical analyses and models by increasing the number of available observations from a wide range of probability distributions (Andréassian et al., 2006; Gupta et al., 2014; Lloyd et al., 2014).

High-quality streamflow measurements are needed for detecting non-stationarity in river flow regimes due to global change.
Long-term streamflow monitoring has enabled data analyses that revealed increasing trends in the frequency of severe floods (Hisdal et al., 2001; Kundzewicz et al., 2013; Blöschl et al., 2019; Gudmundsson et al., 2021; Hannaford et al., 2021), drought



intensification (Vicente-Serrano et al., 2014, 2019; Blauhut et al., 2022), and changes in intermittent river flows regimes (Sauquet et al., 2021). In addition, Meerveld et al. (2020) highlighted inconsistencies between the proportion of potential temporary streams and the occurrence of zero flows reported over 730 gauging stations. Thus, checking flow records before

applying statistical tests is a crucial but delicate task for which there are no common technical guidelines to date, especially for large datasets.

Beyond data analysis, streamflow datasets have been used to set up, calibrate, and evaluate hydrological models through a variety of studies. For example, Chauveau et al. (2013) described the Explore2070 project that used more than 1000 gauging stations to simulate changes in surface water in France by 2065. Forzieri et al. (2014) used a large set of 446 gauging stations to

evaluate their model before addressing the future of streamflow drought characteristics across Europe. de Lavenne et al. (2019) used streamflow time series of 1305 French gauged catchments to evaluate a constrained calibration method of semi-distributed hydrological models.

However, measurements may suffer from flaws that lead to streamflow values that do not reflect the reality (Beven and West-erberg, 2011; McMillan et al., 2012; Wilby et al., 2017). First, instruments are subject to surrounding factors, such as extreme

temperatures or humidity that may alter their functionality. In addition, errors can occur when there is infrequent instrument maintenance because of access issues, for example, which can lead to bias in the measurement. Moreover, streamflow is esti-mated from the conversion of the river water level with respect to a rating curve (Herschy, 2008), which is only valid for given conditions. Thus, measurement errors can occur when the river water level is outside the measurement range of the instrument or the rating curve. Besides, the riverbed may change during extreme events, which can compromise the validity of the rating

curve. Finally, missing data are often filled using interpolation methods during post-processing. Streamflow time series may also include anthropogenic influences (known or unknown to data users), such as agricultural practices with artificial drainage or irrigation, river management with compensation flows, industry with water intakes/releases, and reservoir management for hydroelectric power plants (Wilby et al., 2017).

All these flaws, hereafter called "anomalies" (Leigh et al., 2019), are considered as disinformative data (Beven and West-

erberg, 2011) that may not reflect the natural streamflow dynamics, eventually leading to misinterpretation of hydrological processes. Wright et al. (2015) found that the leverage effect of individual influential data points could substantially influ-ence streamflow predictions depending on the catchment and the model structure. Lamontagne et al. (2013) argue that outliers (i.e., unusually small values compared to the rest of the data) might compromise the accuracy and validity of flood quantile estimation. Therefore, it is crucial to carefully examine the data before trying to make inferences about hydrological processes.

Promising techniques exist for detecting anomalies in water quality and urban water networks (Leigh et al., 2019). These methods often use multivariate analyses that include multiple covariates (when available) or a combination of methods to better detect short- and long-term anomalies in water quality time series (Muxika et al., 2007; Leigh et al., 2019; Rodriguez-Perez et al., 2020) and water level time series (Wiel et al., 2020). However, visual inspection of data is still highly recommended (Barthel et al., 2022) because automatic methods can suffer from misclassification of data points that may require user in-

tervention afterward (Leigh et al., 2019; Wiel et al., 2020). In addition, choosing the most appropriate methods for anomaly detection depends on the goal and expertise of end-users (Leigh et al., 2019). Sebok et al. (2022) used experts' knowledge





to qualitatively evaluate the ability of models to predict future hydrological and climate conditions and concluded that expert elicitation can help to weight and thus decrease the influence of improbable hydrological models. Crochemore et al. (2015) compared numerical criteria of model performance with experts' visual evaluations of hydrographs, and concluded that none of 70 the numerical criteria can fully replace expert judgment when rating hydrographs. Yet, identifying anomalies for a large dataset is time consuming and, as far as we know, no study has focused on the identification of such data in a large dataset of streamflow time series. Therefore, questions remain on their proportion, temporal distribution, and influence on classic hydrological indicators.

This study aims at exploring the results of a large campaign of visual inspection of hundreds of daily streamflow time series 75 in order to detect anomalies: (1) we evaluated the subjectivity of the individual evaluators in terms of the quantity of data reported as anomaly and its agreement with other evaluators; (2) we analyzed the frequency and the temporal changes for each type of anomaly; and (3) we evaluated the influence of the anomalies on the calculation of classic high- and low-flow hydrological indicators. This study provides useful insights into disinformative hydrological data that might help researchers assess the quality of their datasets, in addition to drawing the attention of data producers on how streamflow monitoring could 80 be improved in the future.

## 2 Methods

### 2.1 Data

An initial number of 674 gauged stations were selected in the French HydroPortail database (https://hydro.eaufrance.fr/). The ultimate goal of this catchment set is to serve as a reference streamflow network for the Explore2 project (https://professionnels. 85 ofb.fr/fr/node/1244) that aims to assess the impact of future climate change on water resources in France during the 21st century. Consequently, it is necessary to identify a large set of streamflow stations that provide uninfluenced daily data both for qualifying the natural hydrology and for assessing the performance of daily rainfall–runoff models that are used to simulate future streamflows from climate projections. The criteria of selection of these stations, based on information given in different national databases, were as follows: (1) low and no anthropogenic influence displayed in the database; (2) good data quality 90 for all flow regimes displayed in the database (Leleu et al., 2014); (3) available length of the time series greater than 30 years with less than 10 % of missing data at a daily time step between 1976 and 2019, to mitigate decadal variability; and (4) drained area larger than 64 $km^2$, which corresponds to the area of a pixel from the French SAFRAN reanalysis (Vidal et al., 2010b).

The catchment set covers Metropolitan France well with a variety of hydrological, geological, topographical, and climatic contexts, except for the western part because of the high agricultural influence on streamflow (Figure 1, a). The catchment 95 area of the 674 selected gauged stations ranges from 64 to 111 570 $km^2$ (mean and median of 1278 and 263 $km^2$). The available length of streamflow time series ranges from 26 to 44 years (mean of 39.5 years). The number of available streamflow observations for each day ranges from 503 to 667 stations between 1976 and 2019 (mean of 605 stations, Figure 1, b).





(a)

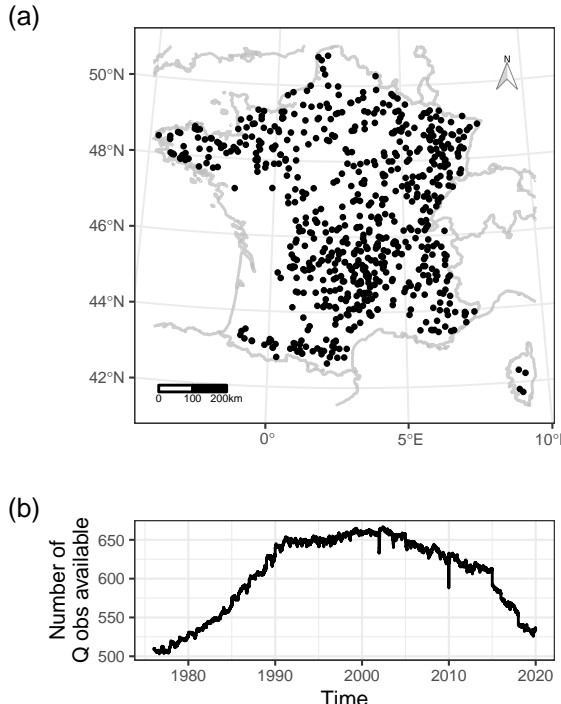

(b)

**Figure 1.** (a) Locations and (b) number of available observed streamflow data (per day) for the 674 gauged stations in France.

## 2.2 Visual inspection of streamflow time series

The Explore2 project focuses on natural streamflow to evaluate the hydrological models. However, the streamflow time series

could be affected by errors or influences, even though they have been marked as having low anthropogenic influence and good quality. Thus, the first objective of the visual inspection of the dataset was to identify stations that were largely influenced by anthropogenic activity (i.e., hydropower production, low-water level support, and reservoir management). The visual inspection involved the participation of 43 evaluators, mostly academic (80 %) and operational (20 %) hydrologists of varying levels of experience (14, 46, and 40 % were considered novice, advanced, and senior, respectively). Each evaluator analysed different

number of time series (later on specified in Figure 3), based on (1) their availability and willingness to analyse a given number of time series, (2) their preference for some hydrographic zones, linked to their expertise, when pertinent, and (3) randomness to attribute the remaining time series. The first step led to the exclusion of 63 stations from the analysis, based on the general feedback given by the evaluators. For these excluded stations, the evaluators mentioned time series that contained too many anomalies to be reported individually, inconsistency of data over several years compared to the rest of the time series, the

absence of any clean summer period, or a too large proportion of missing data filled in by linear interpolation. In addition, data producer helped to identify the potential presence of anthropogenic influences for some of these time series.





The second objective was to visually identify anomalies, i.e., periods when the dynamics of streamflows does not seem natural (e.g., due to dam operation, water withdrawal, instrument failures, unit conversion, or post-processing errors), for the remaining 611 stations. All subsequent analyses were conducted with these 611 stations and the remaining 42 evaluators (since
one of the evaluators only analyzed stations among those that were excluded).

An evaluation protocol was established in order to allow evaluators to report anomalies. All evaluators participated in online meeting sessions whose main objectives were to remind them of the goal of the work and of the evaluation protocol for the time series, to answer any questions about the work, and to enable discussions around any difficulties encountered. Each evaluator had access to a batch of streamflow time series (in html format; see an example in Figure A1) and to a spreadsheet to report
the period and type of anomaly as well as to provide any additional comments that they deemed necessary.

The html file was composed of three dynamic panels displaying the time series of: (1) catchment-aggregated solid and liquid precipitation, and air temperature based on the SAFRAN analysis, (2) streamflow time series in a linear scale, and (3) streamflow time series in a logarithmic scale to highlight potential anomalies for low flows. In order to help identify anomalies, simulated streamflow time series were also displayed, although we emphasized that simulated streamflow could also be flawed
and should not be considered an absolute reference. We used the GR5J lumped rainfall-–runoff model (Pushpalatha et al., 2011) together with the CemaNeige snow accumulation and melt model (Valéry et al., 2014) calibrated on the original time series of streamflows in the airGR package (Coron et al., 2017, 2020) to produce these simulations.

One spreadsheet was provided to each evaluator for every analysis session (usually consisting of a set of 5–10 time series to analyze). This spreadsheet comprised one tab per station that each evaluator had to fill in. The required fields for each
anomaly identified were: its start date, its end date, and the type of anomaly. In addition, the evaluator could provide a general comment on the station. We proposed five types of anomaly, namely, linear interpolation, drops, noise, point anomaly, and other (Figure 2). Linear interpolation consists of periods showing a straight line often due to a filling in of a period with missing data. Noise consists of a periodic pattern in the streamflow time series that may be related to hydro-electricity production or an unknown perturbation in the measurements. Drops consist of a sudden decrease of the measured streamflow that may be due to
water management or technical failures of the instrument. Point anomalies are short-term variations of the streamflow that may be related to the maintenance of the instrument or, for example, to the presence of debris in the river. Evaluators could assign the rating of "other" when an anomaly did not fit any of the four previous types or was a combination of several of them.

We estimated the time needed to evaluate one station to be approximately 10–15 min per evaluator. Since this exercise is subjective, each time series was analyzed by two different evaluators. This allowed a comparison to be made of the subjectivity
in anomaly detection between each evaluator, as well as a better coverage of anomalies in the time series.

## 2.3   Data analysis

### 2.3.1   Combining the feedback

We tested two methods for combining the analyses from the pairs of evaluators for each station: the union and the intersection of the anomalies (Table 1). The union of the anomalies consists of considering a streamflow value as an anomaly in the time





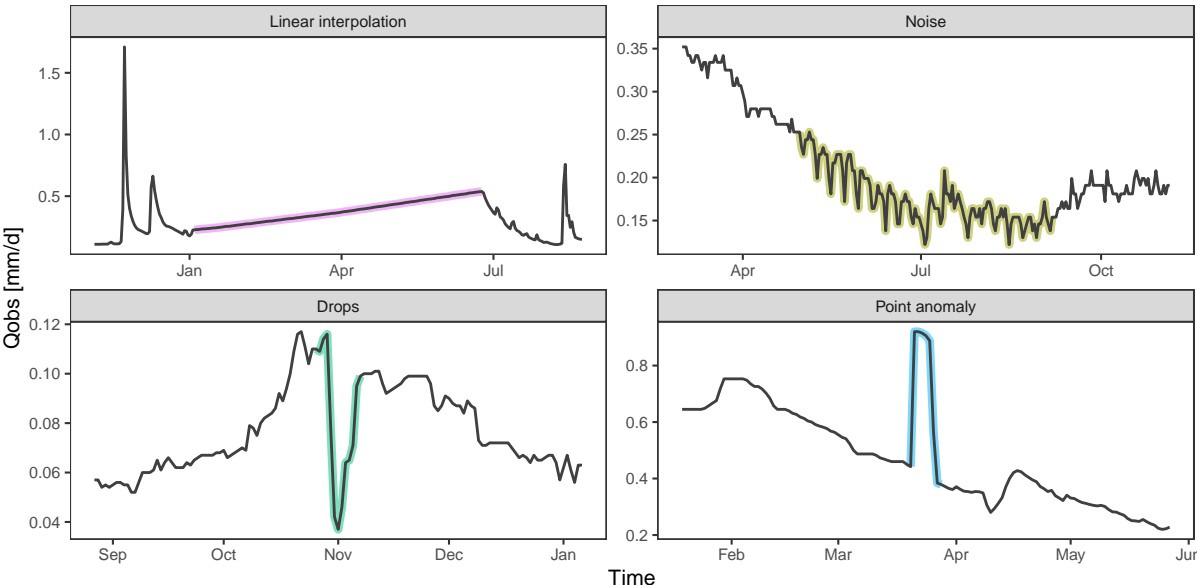

**Figure 2.** Actual examples of the four types of anomalies that could be detected in the dataset.

series if at least one of the evaluators identified it as an anomaly. The type of union of anomaly attributed to a value is either the anomaly type identified by both evaluators if they agree (or if the second type of anomaly is "other", case 1), the anomaly type identified by one evaluator if the other one identified no anomaly (Table 1, cases 3 and 4), or "disagreement" if both evaluators identified different types of anomaly (Table 1, case 2). A value is considered valid if none of the two evaluators identified an anomaly (Table 1, case 5). The pros of using the union of the anomalies is a better confidence in the number of anomalies detected in the time series, at the expense of an increased number of potentially false-positive anomalies detected.

The intersection of the anomalies consists of considering a streamflow value as an anomaly in the time series only if both evaluators identified an anomaly even if the types of anomaly were different (Table 1, cases 1 and 2). A value is considered valid if at least one of the evaluators did not identify an anomaly (Table 1, cases 3–5). The pros of using the intersection of the anomalies is having better confidence in the anomaly of the values identified, at the expense of missing potential anomaly values in the time series (false negative). These two methods for combining feedback will be compared later, and the union method will be ultimately kept for further analyses (see Sect. 3.1).





**Table 1.** Examples of union and intersection of anomalies at a single time step.

| Evaluator 1 | Evaluator 2 | Union | Intersection | Case |
|---|---|---|---|---|
| anomaly A | anomaly A | anomaly A | anomaly A | (1) |
| anomaly B | anomaly A | disagreement | disagreement | (2) |
| anomaly A | - | anomaly A | - | (3) |
| - | anomaly A | anomaly A | - | (4) |
| - | - | - | - | (5) |

### 2.3.2 Analysis by evaluator

We analyzed the individual behavior of each evaluator in terms of sensitivity and agreement in the identification of anomalies with other evaluators. First, we computed the percentage of time identified as an anomaly by each evaluator and by station (Eq. 1) in order to assess the variability among evaluators.

$$P_a = \frac{D_a}{D_Q} \tag{1}$$

where $P_a$ is the percentage of anomaly, $D_a$ and $D_Q$ are the duration (in days) of anomalies and available discharge time series. Second, we estimated the individual agreement of each evaluator with their associated pair when an anomaly was detected. We computed this inter-evaluator agreement for each station as the ratio between the sum of the intersection of anomalies and the union of anomalies (Eq. 2).

$$A_e = \frac{D_a^i}{D_a^u} \tag{2}$$

where $A_e$ is the inter-evaluator agreement, $D_a^i$ and $D_a^u$ are the duration of anomaly considering the union and the intersection of evaluators feedbacks, respectively. The inter-evaluator agreement ranges from 0 % for no agreement at all, to 100 % for a total agreement between the two evaluators. We attributed an inter-evaluator agreement of 100 % if none of the pairs of evaluators found any anomaly in a time series.

### 2.3.3 Analysis by type of error

We compared the inter-evaluator agreement for each type of anomaly by looking at the associated distribution of all anomaly types (i.e., how many times linear interpolation is associated with linear interpolation, noise, drops, point anomaly, other, and none). The proportional distribution of each type of anomaly was computed as the ratio between the length of the union of a





specific type of anomaly and the length of union of all types of anomaly. We also computed the monthly and yearly temporal distributions for each type of union of anomalies.

### 2.3.4 Analysis by station

We assessed the impact of the detected anomalies by computing some hydrological indicators on initial time series and cleaned time series (i.e., time series for which the anomalies were set as missing data). The change rates have been estimated following equation 3 for each time series.

$$C_r = \frac{I_c - I_o}{I_o} \tag{3}$$

where $C_r$ is the change rate of hydrological indicators using initial ($I_o$) and cleaned ($I_c$) discharge time series. We opted for nine hydrological indicators that reflect different parts of the hydrological regime of each river. The seasonal behavior was computed as the monthly mean interannual streamflow. Low-flow periods were evaluated with the 1$^{st}$ and 5$^{th}$ quantiles of the total flow duration curve, the annual minimum monthly flow with a return period of 5 years (noted QMNA5), and the annual minimum of a 30-day moving average of flow with a 5-year return period (noted VCN30$_5$). High flows were evaluated with the 95$^{th}$ and 99$^{th}$ quantiles of the total flow duration curve, and the annual maximum daily flow with a 10-year return period (noted QJXA10).

## 3 Results

### 3.1 Individual behavior of the evaluators

Each evaluator analyzed from 5 to 111 time series, with a mean of 29 stations per evaluator. The percentage of time identified as an anomaly ranges from 0 to 45 % depending on the station and the evaluator, with a median of 0.7 % (Figure 3, a), showing a high variability among evaluators. The median time reported as anomaly ranges between 0.04 and 2.92% according to the different evaluators (Figure 3, a). These median values seem to stabilize around 0.8 ±0.26 % (mean ± standard deviation) for the evaluators who analyzed more than 40 stations in comparison with the other evaluators (0.86 ±0.72 %). The percentages of agreement between evaluators were unexpectedly low, with medians ranging from 0 to 43 % with a mean of 12 % for all the evaluators (Figure 3, b). Regarding the proportion of error identified, the inter-evaluator agreement seems more stable for evaluators who analyzed more than 40 stations (12 % ±4 %) than for the other evaluators (12 % ±10 %).

The low percentage of inter-evaluator agreement was observed for every type of anomaly. Indeed, when an evaluator identified an anomaly, there was no anomaly identified by the second evaluator 59–74 % of the time ("none" type, Figure 4). The higher agreement rates (i.e., both evaluators identified the same type of anomaly) were for the linear interpolation, noise, and drops with 20, 19, and 11 %, respectively. The type of anomaly that was the most often associated with the "other" type was noise (16 %), while the agreement rate for "other" was only 5 %. Even worse, the agreement rate of the point anomaly was the lowest of all types with only 2 % (Figure 4).





**Figure 3.** Individual statistics on (a) the percentage of time reported as an anomaly and (b) the inter-evaluator percentage of agreement for time steps with an anomaly detection. Statistics are displayed by boxplots where the lower and upper hinges correspond to the 25th and 75th quantiles, the vertical bar is the median, the upper and lower whiskers extend up to 1.5 times the interquartile distance from the 25th and 75th quantiles, and the dots are the outliers beyond the end of the whiskers. Each row relates to one evaluator and the number of time series they analyzed. The vertical blue line is the median value considering all evaluators.





Ideally, evaluators would identify roughly the same doubtful periods, and the intersection of anomalies should be recommended to clean up the time series, but the surprisingly low inter-evaluator agreement that we observed would lead to marginal cleaning of the dataset. For this reason, we will focus on the union of the anomalies in the next sections.

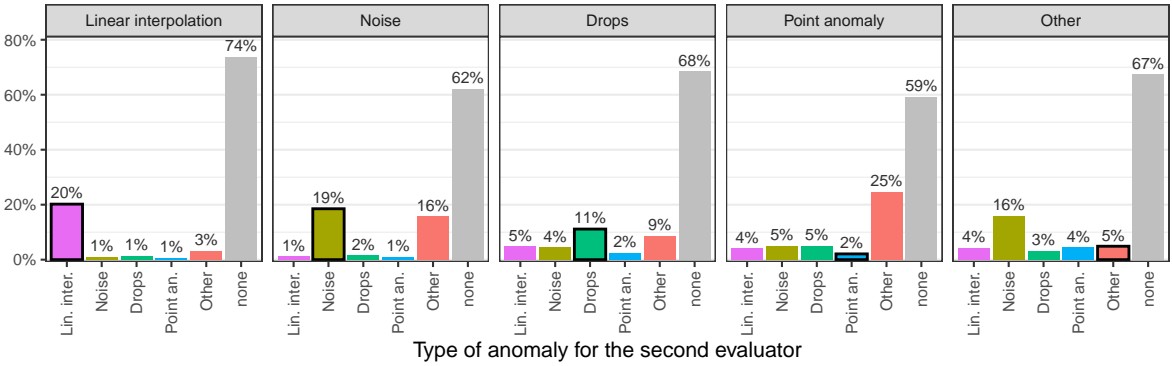

**Figure 4.** Percentage of agreement between evaluators for each type of anomaly. Each bar displays the distribution of the type of anomaly identified by one evaluator when the other evaluator identified the type of anomaly indicated in the bar label. Self-associations (i.e., the same anomaly identified by the two evaluators) are indicated by black-bordered bars (n = 78025, 60108, 21639, 10695, and 59305 days identified by one evaluator as linear interpolation, noise, drops, point anomaly, and other, respectively )

## 3.2 Distribution of the anomalies

The most represented anomalies were the linear interpolations (35 % of the anomalies, Figure 5, a), followed by noise (23 %),
other (20.5 %), disagreement (10.2 %), drops (8.2 %), and point anomaly (3.1 %). The seasonal distribution of the anomalies shows that they occur more frequently during summer than winter months, especially for linear interpolations and noise (Figure 5, b). The long-term evolution of the annual frequency of anomalies seems to decrease from 1976 to 2019 (Figure 5, c), mostly due to a decreasing number of days identified as linear interpolations and noise. A few years showed a higher number of anomalies, such as 1976, 1978, 1985, 1989, and 2003 (relative to surrounding years).

## 3.3 Changes in the length of the time series

The initial length of available data from 1976 to 2019 ranges between 26 and 44 years with a mean and median of 39.6 and 42 years, respectively (Figure 6, a). The average percentage of time series identified as an anomaly ranges between 0 % (for 14 stations) and 46 %, with a mean and median of 2.4 and 1.3 %, respectively (Figure 6, b). Cleaning the time series from these anomalies resulted in a decrease of the mean and median length of the time series by 1 and 1.45 years, respectively. The
length of the clean time series ranges between 23.5 and 44 years, with a mean and median of 38.6 and 40.6 years, respectively (Figure 6, a).



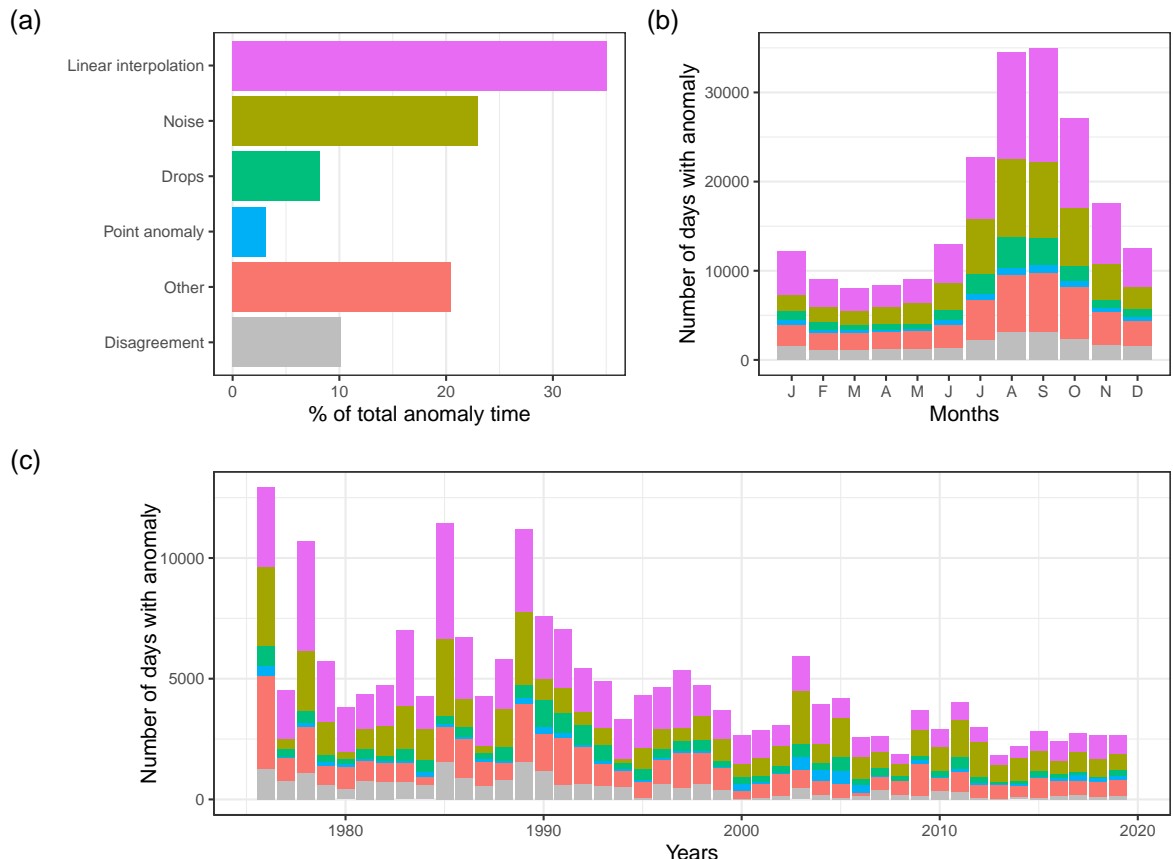

**Figure 5.** Distribution of the types of anomalies (a) in percentage of total time identified as anomaly, (b) with respect to month, and (c) with respect to year for the 611 stations. Each color relates to a type of error such as displayed in (a).

## 3.4 Changes in hydrological indicators

We assessed the impact of removing anomalies from the dataset on hydrological indicators by calculating change rates in hydrological indicators before and after cleaning the time series. Removing the anomalies had little or no impact on the change
rate of high-flow indicator values, while those of low-flow indicators could slightly increase.

The higher impacts were observed for Q1, Q5, and QMNA5 with most (first to third quartiles) of the change rates being between 0 and 6.4 %, 0 and 3.9 %, and 0 and 3.0 %, respectively (Figure 7, a), while a substantial proportion of the stations showed higher change rates for those indicators. Removing the anomalies from the time series had a lower impact on Q50 with most of the changes rates being between 0 and 2.7 %, and a negligible impact on Q95, Q99, QJXA10, and VCN30$_5$ with most
of the change rates lower than 1 %. The change rates of monthly mean streamflow were also negligible overall (Figure 7, b). Most of the stations showed change rates in the monthly mean streamflow from 0 to 1 % in winter months and from 0 to 2.6 %

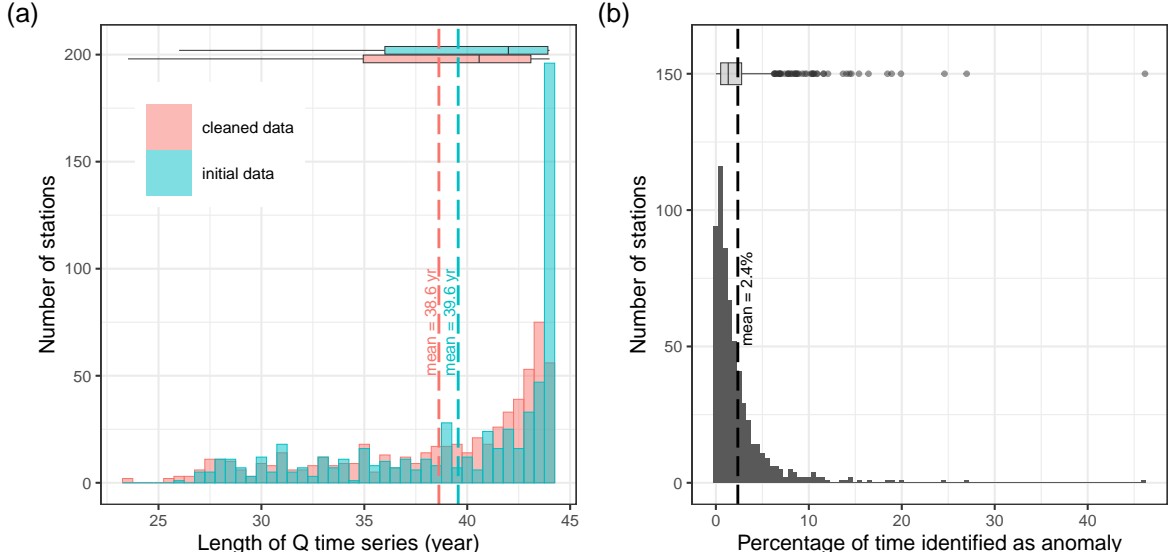

**Figure 6.** Distribution of (a) length of initial observed streamflow time series and cleaned observed streamflow time series, and (b) percentage of time identified as anomalies using the union of anomalies by station. Statistics are displayed by boxplots where the lower and upper hinges correspond to the 25th and 75th quantiles, the vertical bar is the median, the upper and lower whiskers extend up to 1.5 times the interquartile distance from the 25th and 75th quantiles, and the dots are the outliers beyond the end of the whiskers. The dashed blue line relates to the mean of each population.

during summer months, i.e., the monthly mean streamflow increased by less than 2.6 % after we removed the anomalies from the time series.

## 4 Discussion

### 4.1 Subjectivity of the evaluators

One of the main results of our study is the high subjectivity in detecting anomalies, as reflected by the variability in the percentage of time identified as anomaly, as well as in the low inter-agreement on the timing and on the type of anomaly identified among evaluators.

The high variability in the percentage of time identified as anomaly (Figure 3, a) might reflect the quality of the time series that the evaluators had to analyze, which may be heterogeneous among the 611 stations. Another explanation could be that this variability reflects differences in the level of expectations of the evaluators, therefore suggesting that the individual subjectivity is high during visual inspection of streamflow time series. This variability seems to stabilize for the evaluators who analyzed more than 40 stations, which raises two questions: Is it related to the variability of the quality of the station time series that



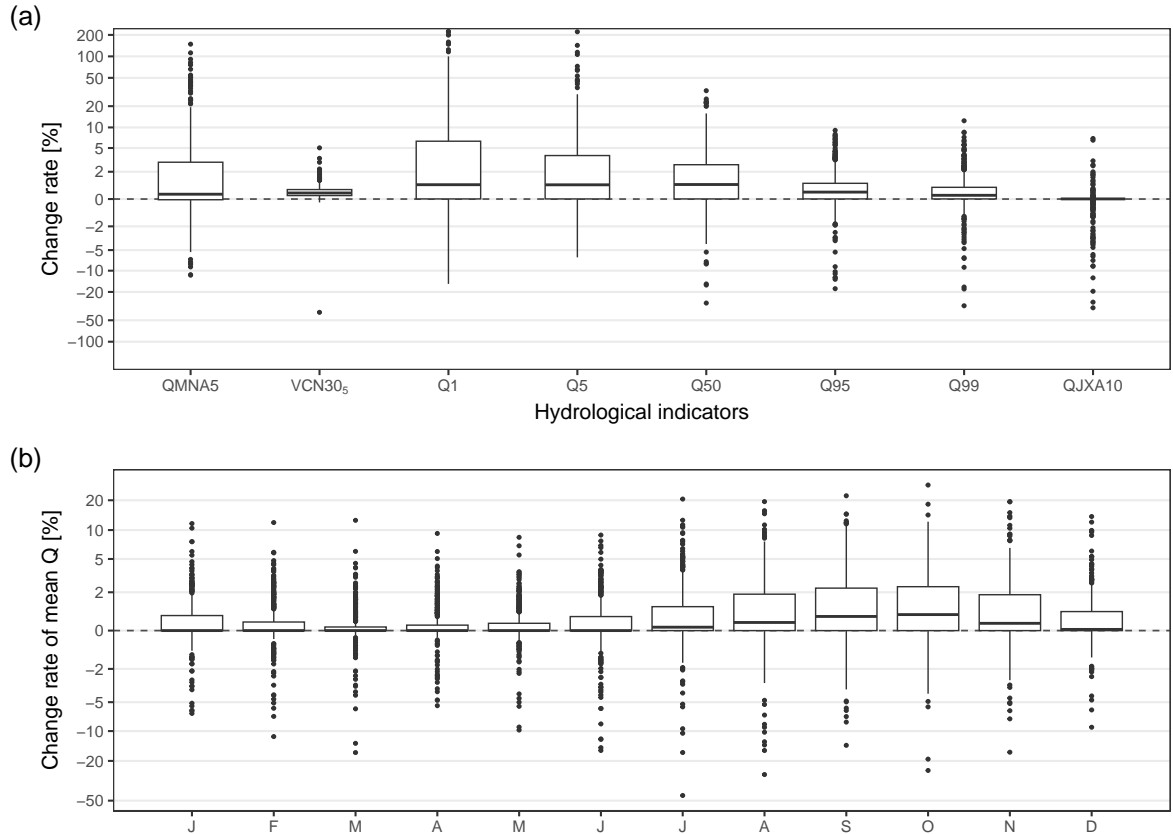

**Figure 7.** Change rates of hydrological indicators after cleaning the anomalies from the streamflow time series relative to indicators calculated with the initial streamflow time series. (a) is the distribution of the change rates for quantiles (1, 5, 50, 95, and 99[th]), low-flow indicators (QMNA5 and VCN30$_5$), and a high-flow indicator (QJXA10), and (b) is the distribution of the change rates of monthly mean streamflows. Statistics are displayed by boxplots where the lower and upper hinges correspond to the 25[th] and 75[th] quantiles, the vertical bar is the median, the upper and lower whiskers extend up to 1.5 times the interquartile distance from the 25[th] and 75[th] quantiles, and the dots are the outliers beyond the end of the whiskers.

lowers over 40 stations analyzed? Did the evaluators gain experience in visual inspection during the operation, so that their

expertise converged toward a value similar to that of other evaluators?

The low overall values of inter-evaluator agreement (Figure 3, b) reflect the high subjectivity in evaluators that makes the identification of anomalies highly challenging. This is true for the identification of the timing, but also for the attribution of a type of anomaly as shown by the high proportions of "other" and disagreement types of anomaly identified (Figure 5, a), and by the low percentage of self-association of type of anomaly (Figure 4). It seems that each evaluator has their own intuition of

what is a proper time series of streamflow and what type of anomaly is identified. This subjectivity might reflect the variety of





expertise or the level of expectations of hydrologists. One might be more focused on flood events, thus looking more at high-flow periods. Another might be more interested in groundwater flows and the seasonality of the streamflow dynamics. And yet another might be more interested in droughts, thus looking for the low-flow dynamics. The subjectivity of hydrologists has also been observed during visual evaluations of model performance (Alexandrov et al., 2011; Crochemore et al., 2015; Melsen, 255 2022) and of groundwater hydrographs (Barthel et al., 2022).

Some anomalies appeared easier to identify than others, such as linear interpolation and noise (Figure 5, a). One reason that may explain why linear interpolation and noise were the most represented anomalies is that they can last for weeks or months, while point anomaly and drops may last a few days (mean length of 22 and 73 days for linear interpolation and noise, respectively), which should also make them easier to detect, even though the inter-evaluator agreement for linear interpolation 260 was only 20 % (Figure 4).

## 4.2 Temporal distribution and impacts of the anomalies

The annual frequency of anomalies showed a long-term decrease (Figure 5, c) that is not related to the number of available data (see Figure A2 for the proportion of anomaly by year). Thus, it may reflect an overall improvement of the streamflow monitoring techniques or more efficient data post-processing thanks to software improvements. Another option is that the data 265 supplier did not have the means to thoroughly verify old data that might be inherited from other services or from old records. However, the long-term decrease in anomalies might also be explained by the greater focus of evaluators on streamflow analysis in the beginning of the time series, which is supported by the decreasing number of disagreements after 2000 (Figure 5, c), depicting a behavior where evaluators only picked the most obvious anomalies. The few years showing a higher number of anomalies (1976, 1978, 1985, 1989, and 2003, Figure 5, c) reflect some climatic events in France, such as the heatwaves and 270 droughts in 1976 and 2003, which may have affected the proper functioning of the instruments. Indeed, Vidal et al. (2010a) identified all these years as major drought events related to exceptional precipitation or soil moisture deficits.

Beside, the temporal distribution of anomaly showed a seasonal pattern (Figure 5, b) that can explain by many factors. The summer months are characterized by low flows, which are harder to monitor than the medium-to-high flows because of the resolution of the flow measurement instruments. The summer period is also traditionally a holiday period in France, meaning 275 that instruments can be fixed less promptly. In addition, low-flow periods are proportionally more impacted by anthropogenic influences such as water withdrawal, energy production, or water releases. Another reason might be the use of the logarithmic scale to plot the streamflow panel of the html files provided to the evaluators. This scale might have disproportionately magnified some anomalies that were minor in reality.

Our results show that the mean proportion of time series identified as anomaly remains relatively low (Figure 6), even though 280 we excluded the union of anomaly reported by evaluators. Excluding these periods from the initial dataset had little influence on the length of available data for most of the stations. It should be noted that these data were previously labeled as good-quality data in the metadata provided by the producer. This means that the time series had been considered as valid before they were made available in the database, which may explain the low percentage of data identified as anomaly. However, since the



anomalies were more frequently reported during the summer months (Figure 5, b), their removal from the time series might
induce a seasonal bias that may impact the calculation of hydrological indicators.

Indeed, the change rates were higher for hydrological indicators from July to November and for Q1, Q5, and QMNA5
(Figure 7), which could have implications from an operational point of view. In addition, the rare stations with a negative
change rate suggests that lower flows were more often identified as anomaly (Figure 7). These results suggest that wet periods
were better monitored overall than dry periods or that anomalies were easier to detect during low-flow periods because of
the logarithmic scale on the streamflow time series we provided to the evaluators. Some stations displayed as outliers (dots
in Figure 7) showed changes that are nonetheless larger than described above. These higher change rates might be related to
specific cases in our sample of stations, or to the proportion of data removed from the time series. Indeed, a low initial value of
streamflow or few remaining data after removing the anomalies from the time series can have a large impact on hydrological
indicators, especially for lower quantiles such as Q1 and Q5, and for calendar-constrained indicators such as QMNA5.

## 295 5 Lessons and perspectives

This section provides feedback on the design of the visual inspection campaign and some considerations about how such a
campaign could be used to improve the use of streamflow datasets. These comments are intended for data producers, data
users, and anyone interested in repeating a similar visual inspection campaign of streamflow time series.

### 5.1 Considerations for data producers

The number of available streamflow data in France is huge and of good quality most of the time, as shown by the low proportion
of anomalies found in the time series (Figure 6). The data provided over large spatial and temporal scales at high resolution are
essential to build and test hypotheses on hydrological processes or to estimate catchment indicators. We noticed an improve-
ment in river monitoring in terms of quantity as the available data increased from 1976 to 1990, but also potentially in terms of
quality as anomaly rates decreased from 1976 on. Nevertheless, we observed that available data have been decreasing slightly
since 2005 and sharply since 2015 (Figure 1, b). The latter decrease might be due to a delay between streamflow monitoring
and publishing in the database; nonetheless, we are concerned about the slight decrease, which seems to be related to a gradual
closure of gauged stations. Crochemore et al. (2020) also observed a decrease in the availability of data worldwide for politi-
cal, economic, or privacy reasons. We would like to emphasize that long-term data are essential for hydrologists, especially for
those studying the effect of climate or land use change on the water cycle. In addition, we noticed that medium- to high-flow
events appear to be well monitored, perhaps due to the aggregation of measured data to daily discharge; however, the higher
rates of anomaly during the drought periods and drought years such as 1976 and 2003 (Figure 5) showed that an improvement
of the techniques of low-flow monitoring is still possible (Horner et al., 2022). Such an improvement would be very valuable
to the hydrological community (Meerveld et al., 2020).





## 5.2 Considerations for data users

Data analysts and modelers are aware that perfect time series do not exist for many reasons (Wilby et al., 2017), thus pre-processing is mandatory before any use of the data. One of the main findings of this study is that data cleaning is highly subjective, as shown by the large proportion of heterogeneity in the temporality and types of anomalies that were reported by the evaluators (Figure 3 and 4). This high subjectivity raises questions on the best practices for data cleaning, such as how many evaluators should be involved in the detection of anomalies and how their feedback should be combined. For example, 320 we could picture a process that combines the anomalies coming from three or more evaluators and consider as actual anomalies the periods for which at least two evaluators agree.

Our results show that the union of anomalies of two evaluators had a marginal effect on high-flow hydrological indicators, but could impact the low-flow indicators such as Q1 and QMNA5, which could have implications for water management regulations. Low impacts were observed for most of the stations of the large dataset we used (611 stations), but we emphasize 325 that the impact of data cleaning on individual stations might still be high for particular cases and the awareness of data users should increase as the size of the dataset decreases. We did not find any clear relationship between change rates in low-flow hydrological indicators and the proportion of data removed, while the absolute change rates in Q50, Q95, and Q99 seem to increase with the proportion of data removed from the time series (see Figure A3). van den Tillaart et al. (2013) and Brigode et al. (2015) showed that systematic errors, outdated rating curves, or a single extreme event (e.g., extreme flood) could affect 330 model performance and parameter estimation. Thébault et al. (under review) reported that artificially corrupting time series has little effect on model calibration over a large dataset. Nevertheless, Perrin et al. (2007) and Ayzel and Heistermann (2021) showed that modelers can remove a large proportion of data from the calibration set, such as potential anomalies, for example, without compromising the estimates of the model parameters, as long as the calibration period covers dry and wet periods.

## 5.3 Feedback on the visual inspection campaign

This section aims at providing suggestions to those interested in reproducing such a campaign of visual inspection of streamflow time series. The objective of our campaign was to analyze a large dataset of streamflow time series in order to remove anomalies that could impact the evaluation of hydrological models. The dataset comes from a wide range of catchment conditions with different responses to rainfall events, hence the need to simultaneously compare temperature, precipitation, and streamflow time series. The first suggestion is that the number of anomaly types provided should be as low as possible to avoid 340 confusion. Indeed, our results suggest potential confusion between the drops and point anomaly types (Figure 4). If the types of anomaly are not well defined, each evaluator can picture different kinds of anomalies for a given period, or may consider that the proposed types share similarities, which makes it difficult to choose one type or another. A phase of inter-calibration of evaluators, and even better with the data producer when possible, is highly recommended as it could reduce the subjectivity of such an exercise. Furthermore, we suggest adding a confidence rate in addition to the periods and type of anomaly, which 345 would encourage evaluators to identify more doubtful periods, therefore increasing the intersection of identified anomalies.





Including the confidence rate in a study such as ours might also make it possible to deepen the investigation on the subjectivity of each evaluator.

One important feature of visual inspection is the figure layout of the time series provided to the evaluators, as also noted by Barthel et al. (2022). Indeed, using a logarithmic scale for the streamflow axis might have facilitated the identification of anomalies in low-flow conditions, thus it may have artificially increased the anomaly frequency during low-flow periods (Figure 5). In addition, evaluators seemed to lose focus when analyzing long time series (44 years). Consequently, one should consider splitting streamflow time series to investigate and perhaps avoid this potential effect of weariness, or to simplify the anomaly-reporting procedure with an interface that allows one to select the period and type with a mouse click.

An automatic detection of anomalies could avoid these issues of subjectivity and weariness. Using the bias between model simulations and measured time series could be a starting point for identifying potential anomalies. Unfortunately, to our knowledge, these techniques still require improvements. Such an algorithm should be flexible regarding the types of anomaly to identify, and might be trained for each study to avoid the risk of removing data of interest (e.g., using a visual inspection such as the one reported in this study). Ideally, hydrologists should share a common library of anomaly types such as suggested by Wilby et al. (2017). Promising perspectives in anomaly detection could be using covariates, such as biogeochemistry, conductivity, or water temperature time series in addition of streamflow, since they may reflect the hydrological conditions and flow paths, or using paired catchment and comparing their double-mass curves of discharge time series.

## 6 Conclusions

This study explored the results of a large visual inspection campaign of streamflow time series in France that aimed at detecting non-natural records. The objectives of this study were to evaluate the subjectivity of evaluators in detecting anomalies, to analyze the frequency and temporal distributions of anomalies, and to assess their impact on commonly used hydrological indicators.

Each of the 611 time series was visually inspected by 2 out of 42 evaluators in order to report doubtful periods such as linear interpolation, noise, drops, point anomaly, or other. The feedback was combined using the union and the intersection of the periods of anomaly reported.

The main result of this study is the high subjectivity of the evaluators' behavior in detecting anomalies. Indeed, the surprisingly low inter-evaluator agreement rates (mean value of 12 %) reflect the variety of feedback by the hydrologists. Evaluators more frequently reported periods of anomaly during summer, which could be related to low flows that are more difficult to monitor than high flows, but also to relatively higher anthropogenic influences during this period. Consequently, we observed higher impacts of the anomalies when analyzing low-flow indicators calculated on the initial time series and on time series where the anomalies were removed, although the change rates remained low overall, below 5 % for most of the time series in our large dataset.





*Data availability.* Observed streamflow data are available from the French HydroPortail database (http://www.hydro.eaufrance.fr/, last access: 1$^{st}$ September 2021). The information on the time steps reported as anomalies and the type of anomalies can be found at: https://doi.org/10.57745/SO2WOV (Strohmenger and Thirel, 2023).



380    **Appendix A**

**Figure A1.** Example of an html file displaying streamflow time series that was sent to evaluators.





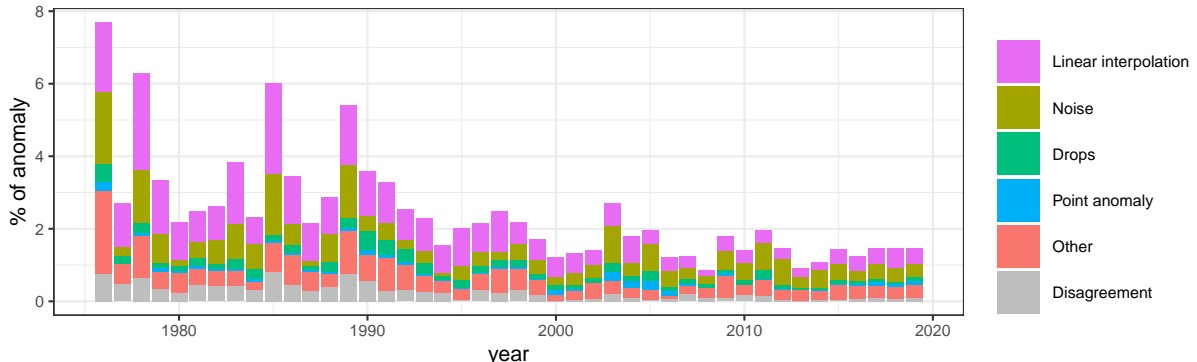

**Figure A2.** Proportion of anomalies (number of anomalies / number of available Qobs) with respect to year for the 611 stations.





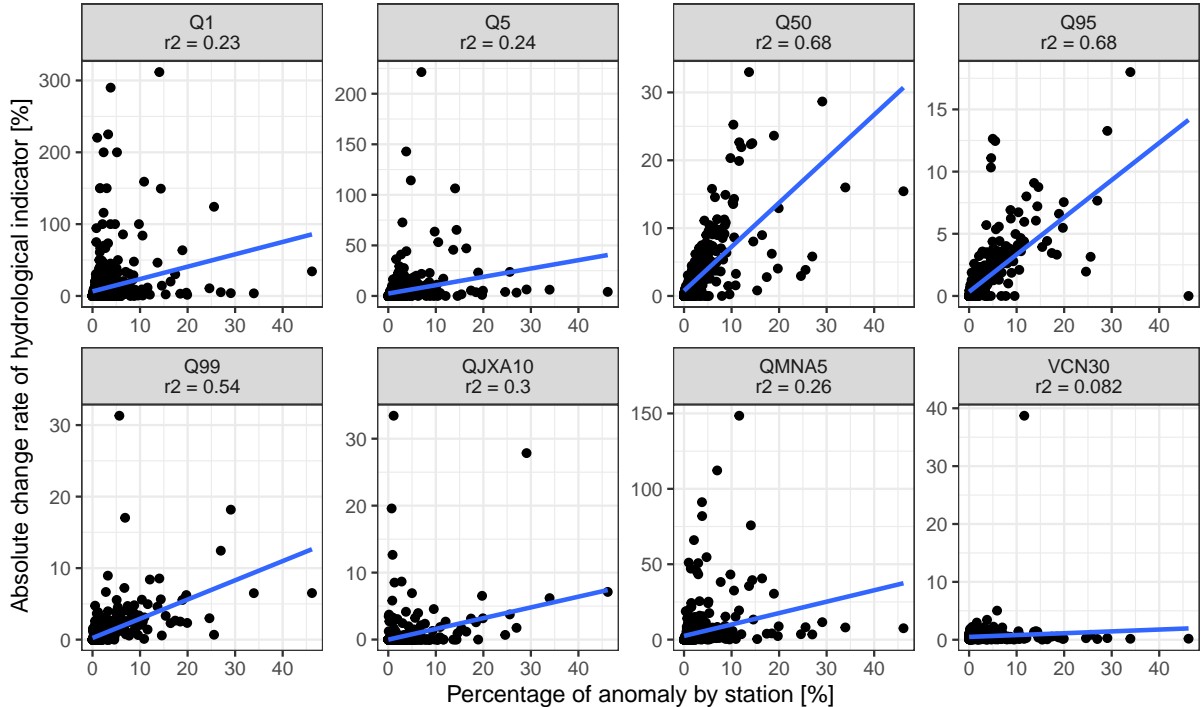

**Figure A3.** Absolute change rates of hydrological indicator vs. the proportion of anomalies for the 611 stations.





*Author contributions.* LS, GT, and CP conceived the evaluation framework; GT prepared the html files (Delaigue et al., 2020); LS and GT organized the online meetings; LS analyzed the feedback and drafted the article; all authors analyzed the time series and reviewed the article.

*Competing interests.* The authors declare that they have no conflict of interest.

*Acknowledgements.* The Explore2 project funded LS's postdoctoral position. We thank the additional evaluators, Patrick Arnaud, Audrey
385 Bayle, François Bourgin, Yvan Caballero, François Colleoni, Joël Gailhard, Thibault Hallouin, Enola Henrotin, Baptiste Lévêque, Arlette Robert-Vassy, Paul Royer-Gaspard, Michael Savary, and Gaëlle Tallec, who participated in the visual analysis campaign. The French government (SCHAPI) and the HydroPortail database are acknowledged for providing hydrological data.



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
