# Peer review of "On the visual detection of non-natural records in streamflow time series: challenges and impacts"

_Hydrology and Earth System Sciences, 2023_

## Referee Comment (RC3)

**Referee Comment of Alexander Gelfan on Strohmenger et al. "On the visual detection of non-natural records in streamflow time series: challenges and impacts"**

The study is a new attempt to reveal non-natural records of different origins, including erroneous ones, in streamflow time-series. The authors developed a comprehensive protocol for visual inspection of river flow data and involved 43 experts to detect anomalies in 674 streamflow time series in France using the protocol. The study showed a huge variability in the assessments of experts and confirmed the prevailing a priori ideas about the predominance of subjective factors when deciding on the presence of anomalies. Nevertheless, even with such uncertain results, the authors were able to formulate several recommendations, among which two seem to me to be the most important: (1) analyze as few types of anomalies as possible; and (2) allow experts to supplement the detected anomalies with confidence estimates.

Overall, I believe that the manuscript addresses relevant scientific issues and contains results that could make a useful contribution to future studies. The scientific methods and assumptions are valid and clearly outlined. The presentation is well structured and clear. I find the study to be interesting and recommend the manuscript for publication after minor revisions.

Compared to Martin Gauch's excellent review already published, there is very little I could add. I fully agree with the major comments 2, 3, and 5 of this review; namely, following these comments, I also recommend the authors: to compare the obtained "change rates" with the values that would have been obtained by randomly deleting the same amount of data from the analyzed series; to evaluate the inter-evaluator agreement within certain categories of experts; and to assess whether the quality of hydrological simulations will change when evaluating the performance criterion on the cleaned series.

In addition to the technical comments below, I would like to make two more general notes, and I'll be grateful if the authors comment on these issues in their response.

The first one concerns to the organization of the related studies. It seems logical to me to make one preparation. Before the main study begins, ask experts to weigh in on one or a few (but not many) reference streamflow time-series where some of the data has been substituted with fictitious data that the organizers are aware of. This stage will provide a preliminary general sense of the potential levels of expert agreement and the accuracy of their expert judgments.

The second general comment relates to my personal view on the perspective of visual detection of anomalies in the streamflow time-series. Given the inevitable high level of subjectivity in expert judgments (associated, first of all, with the experts' experience), I believe that expert assessments would become more effective if not the entire series of observations were subjected to visual analysis but only its suspicious parts, previously identified using popular quantitative

algorithms (k-nearest neighbors, clustering based algorithms, machine learning algorithms, etc.). This will make it possible to reduce subjectivity and increase the information content of expert analysis.

**Technical comments**

Line 90: "available length of the time series greater than 25 years…" as it follows from line 96

Line 138: It is unclear to me what the reason was to limit an evaluation time. It seems to me that it is more important to get a thoughtful assessment than a quick response.

Line 167: "…are the duration of anomaly considering the intersection and the union…, respectively."

Fig. 3b: It is not entirely clear how the inter-evaluator agreement between an expert who analyzed data from 111 stations and another expert who processed data from a much smaller number of stations (say, 10) was established. Please clarify

I suggest including the main recommendations formulated in subsection 5.3 and related to visual inspection of streamflow time series into the conclusions.

---

## Author Comment (AC1)

Thank you very much Dr Gauch your review. We provide below answers to the reviewer's comments. Reviewers' comments are shown in black. Authors' responses are shown in green.

During the review process, we identified an error in the computation of one of the hydrological indicators ($VCN30_5$). We corrected the error. The change rate values of the $VCN30_5$ are now more consistent with those of the QMNA (both low-flow indicators) than before. We will update the figure 7 and the values of changes rate of the $VCN30_5$ in the results section of the manuscript. We will also update the appendix C.

**Summary**

The paper describes the outcomes of an effort to have human annotators detect non-natural/incorrect streamflow measurements. The authors find remarkable inconsistency in different annotators' responses.

Overall, I think this paper is well-written and to the point. I appreciated the discussion and suggestions for future related studies.

I think that a few changes in the analysis of results could further improve the paper. Please find my detailed comments below.

Thank you for your kind words about the article, and for all your constructive suggestions.

**Major Comments**

1.  I would guess that it is not always clear which exact timesteps around an anomaly should be annotated. I think it might therefore be interesting to see if the consistency increases noticeably when we consider a windowed approach: e.g., consider two annotations to agree when they fall within a 1-week window.

Indeed, the inter-evaluators agreement might increase by including more flexibility in the start and end dates reported by the evaluators. We tried windows of plus and minus 3 and 7 days for allowing the intersection of anomalies, and then we computed the inter-evaluator agreement. The overall median of agreement rate increased to 15 % (+/- 3 days) and 17.4 % (+/- 7 days) vs. 13 % in the manuscript (+/- 0 days). Although the inter-evaluator agreement increases with these extended windows, it remains surprisingly low. We propose to add a sentence about this experiment in the discussion of the results in section 4.1:

*"This is true for the identification of the timing of anomalies, although including a margin of three or seven days on the start and end date slightly increased the inter-evaluator agreement to 15 and 17 %, respectively. The subjectivity of evaluators also reflects in the attribution of a type of anomaly as shown by the high proportions of "other" and disagreement types of anomaly identified (Figure 5, a), and by the low percentage of self-association of type of anomaly (Figure 4)."*

That being said, choosing the adequate length of the windows around the time steps is tricky because of the duration of the different types of anomaly (linear interpolation and noise are longer than point anomaly, several weeks vs. few days).

2. I appreciate the effort to measure the impact of removing annotations through the "change rate", but it is hard to put the values into perspective. For that, I would propose the following baseline: report the change rate that you get when randomly removing the same fraction of time steps (ideally, do this a few times and report the average change rate).

We agree that such a baseline could help interpreting the changes rates of the hydrological indicators. We computed them, following your suggestions, for each station:

   a. Count the number of days reported as anomalies (n) in a time series

   b. Attribute anomaly to n random time steps for which discharge was observed

   c. Compute the hydrological indicators for this randomly altered time series

   d. Repeat this 100 times by time series and average the change rates

We plotted the results using the same layout as in the manuscript (figure 1 below, which complements figure 7 of the manuscript). Results show that randomly removing time steps from streamflow time series (see the baseline boxplots) has negligible impacts on hydrological indicators, much lower than those observed when removing actual anomalies reported by the evaluators.

[Figure]

Figure 1: Change rates of hydrological indicators after cleaning the anomalies from the streamflow time series relative to indicators calculated with the initial streamflow time series. (a) is the distribution of the change rates for quantiles (1, 5, 50, 95, and 99th), low-flow indicators (QMNA5 and VCN305), and a high-flow indicator (QJXA10), and (b) is the distribution of the change rates of monthly mean streamflows. Colors refer to the type of experiment:

actual (black) relates to changes rates when removing actual anomalies reported by evaluators, baseline (blue) refers to randomly removed time step from the time series. Statistics are displayed by boxplots where the lower and upper hinges correspond to the 25th and 75th quantiles, the vertical bar is the median, the upper and lower whiskers extend up to 1.5 times the interquartile distance from the 25th and 75th quantiles, and the dots are the outliers beyond the end of the whiskers.

We propose to mention this in the manuscript in section 4.2.

*"Indeed, the change rates were higher for hydrological indicators from July to November and for Q1, Q5, and QMNA5 (Figure 7), which could have implications from an operational point of view. We assessed the impact of removing random time steps from streamflow time series in the same proportion than reported by evaluators (results not shown). We observed that the change rates for these indicators when removing anomalies were larger when using evaluators' feedbacks than when removing random time steps."*

3. It might be interesting to see if there is a noticeable difference in the responses/consistency of novice vs. advanced vs. senior annotators.

This is an interesting suggestion. We checked the relationship between the subjective statistics (percentages of anomaly reported and inter-evaluator agreement) with the level of experience of the evaluators. The percentage of anomaly reported seems to decrease as the experience of the evaluator increases; however, the variability between individuals is too high to conclude for a strong relationship (Figure 2). One interpretation could be that senior evaluators tend to neglect anomalies they believe will not affect a hydrological analysis, while novice evaluators tend to be more exhaustive and try not to miss any potential anomaly.

For this experiment, we refined the level of experience of the evaluators in our database. Thus, the proportion of each level of experience of evaluators slightly changed from 14 to 24% for novice, and 46 to 38 %, for advanced and senior (see L104). This will be adjusted and additional analysis will be mentioned in section 4.1.

[Figure]

Figure 2: Percentage of time reported as anomaly (left) and inter-evaluator agreement (right) with respect to the level of experience of the evaluators. The height of the column and the error bar indicate the mean and the standard deviation within the groups, respectively.

4. Reproducibility/data availability: For purposes of reproducibility, I would appreciate the download to include:
   a. the streamflow data (and ideally also the supplemental information like precipitation, temperature, GR5J simulation, etc.), such that others can reproduce and build upon the analyses. The French-only HydroPortail link is not helpful and it's unclear what data one would need to collect from there.
   b. the code used in the study (e.g., to calculate the various statistics)
   c. information on the evaluators (background academic/operational, level of experience)

Unfortunately, we are not allowed to provide for download data that we did not produce and that we do not own.

Regarding hydrological data, we would like to mention the existence of the hub'eau API to collect streamflow time series over France from the HydroPortail. It is a French service that aim at simplifying access to water data:

https://hubeau.eaufrance.fr/page/api-hydrometrie

A tutorial is available here (unfortunately also in French, but translation tools seem to be efficient in providing an English version):

https://hubeau.eaufrance.fr/page/api-qualite-cours-deau-tuto

We will mention the hub'eau API in the data availability section.

Regarding the SAFRAN meteorological data we used, we are also not allowed to share it. However, the data producer, Météo-France, generally provides these data for research purposes. In the present case, to run the GR hydrological models the use of SAFRAN is not

compulsory: any open access European dataset (e.g. ERA-Interim) could have been used, as the performance of the model is not the core of this work and we expect larger datasets still to have consistent precipitation signals for the evaluation phase. We only used SAFRAN because this is the best dataset at our disposal and usually use it in most of our studies over France.

Based on the fact that we are not allowed to provide neither hydrological not meteorological data, we prefer not to provide GR5J simulations. However, the GR5J model is available in an open access mode in airGR (https://cran.r-project.org/web/packages/airGR/index.html) and could be used by anyone to reproduce the experiment.

We provided additional information on the background and the level of the evaluators in a new file (evaluators_profile_Explore2) available on the data deposit: https://entrepot.recherche.data.gouv.fr/dataset.xhtml?persistentId=doi:10.57745/SO2WOV

Thanks for the suggestion; we apologize for not being able to positively answer to the entirety of this demand.

5. It would be really interesting to see how removing bad data impacts the quality of hydrological models. Is a model that is calibrated/trained only on "good" timesteps better than one that is calibrated on all timesteps (on a validation period)? This might be out of scope for this paper, but could at least be mentioned as future work.

Absolutely! Actually, we are currently working on the impact of anomalies on models parametrization and simulations. However, in our opinion, this is a large topic that would merit a whole article. We will add a sentence to discuss this perspective in the discussion section (see also comment on L328ff below)

**Minor Comments**

L90 states >30 years of available records as a condition, but then L96 reports that some rivers have only 26 years of streamflow. Am I missing something?

L90 states "30 years with less than 10 % of missing data" which should be >= 27 years. You are right; one of the station is actually shorter (26 years) due to a human mistake during the station selection stage.

We will rephrase, as also suggested by RC3, for:

*"(3) available length of the time series greater than 26 years at a daily time step between 1976 and 2019"*

L224 "little impact on the change rate". If I understand correctly, the change rate is exactly what measures the magnitude of impact. I.e., the change rate only exists after removing the anomalies (and not before). So shouldn't this rather be reworded to something like "did not result in large change rates"?

Change rates were computed as the difference between hydrologic indicators with and without anomalies included in streamflow time series. You are right, "removing anomalies" do

not affect change rates, but rather "results in" change rate modification. We can rephrase the sentence as follows:

*"Removing the anomalies had little or no impact on the high-flow indicator values, while those of low-flow indicators could slightly increase."*

L242f Regarding the first reason you provide for the stabilized variability: I don't think it is the variability that lowers as the number of stations increases. I think it is the quality of the estimate that improves as more samples are available. Both station quality and human randomness have some variance, which will be better estimated from larger sample sizes.

You've said it better than we could. We propose to rephrase as follows:

*"This variability seems to stabilize for the evaluators who analyzed more than 40 stations, which raises two questions: Is it related to the variability of the quality of the station time series that is better estimated over 40 stations analyzed? Did the evaluators gain experience in visual inspection during the operation, so that their expertise converged toward a value similar to that of other evaluators?"*

L266: This notion of annotators getting "bored" could be verified by checking whether the number of annotations also decreases across stations (those rated first vs. those rated last)

True, however, we cannot retrieve the information on when evaluators examined the time series, thus such analyze cannot be performed.

L281: Do you know how this "good-quality" label was assigned? Does this mean another human had visually deemed these records as good? Or via automatic checks?

Quality label was assigned by data producers and describes the quality of measurements during low-, mid- and high-flow periods. Each data producers has its own criteria to validate the data, based on daily, weekly or monthly expertise, statistical analyses, and hydrometric monitoring software. A good quality label means that the data producer removed the known period of the streamflow measurement disturbance such as for example sensors maintenance operations or sensor drift.

L290: The logarithmic scale is an important detail to me, because I think it makes linear interpolation much harder to notice. This might be something to add to L349f.

We feel that streamflow time series variability is so high (at short time scale) that any linear interpolation longer than a few days is easy to notice, even in logarithmic scale. However, we strongly encourage to always display time series using a linear scales at least, and we provided observed time series both using a log and a linear scale.

L328ff: I would move this brief literature review to the related work section and remove it from the discussion.

We provided this literature review to introduce model's sensitivity to potential anomalies. As you mentioned in a previous comment (comment 5), we feel that this is out of scope for this paper, and then we would like to keep theses sentences in the discussion section. However, we did not clearly state it in the manuscript. We will add a sentence to mention this perspective in the discussion section.

*"Hydrological models may also be influenced by anomalies, and further studies should investigate how parameters and simulated streamflow change when removing anomalies from the training period data. "*

Out of curiosity: would a map of France show any spatial patterns in the annotations?

We did plot the map of anomaly rates, as we expected to see the propagation of anomalies from upstream to downstream stations, but we did not find any spatial pattern or spatial correlation of the anomaly.

**Typos**
L125 rainfall—runoff

This will be corrected, thanks.

---

## Author Comment (AC2)

Thank you very much Dr Kratzert for your comments.

Reviewers' comments are shown in black. Authors' responses are shown in green

During the review process, we identified an error in the computation of one of the hydrological indicators ($VCN30_5$). We corrected the error. The change rate values of the VCN30 are now more consistent with those of the QMNA (both low-flow indicators) than before. We will update the figure 7 and the values of changes rate of the $VCN30_5$ in the results section of the manuscript. We will also update the appendix C.

The manuscript, by Laurent Strohmenger et al., presents results of a survey, in which 42 hydrologists were asked to annotate anomalies in streamflow time series. In my eyes, the main results of this manuscript:

1. A dataset of anomalies in streamflow time series annotated by human experts.
2. Another proof of the subjectivity and inconsistency of human experts, when tasked to rate/compare/annotate hydrographs.
3. Recommendations for future studies of this kind.

Overall, the manuscript is very well written, easy to follow. Given the already published comments by Martin Gauch, I have very little (see below) to add and recommend the publication of this manuscript after considering these minor comments.

**Data sharing.**
The authors state in L 354ff. The following:

*An automatic detection of anomalies could avoid these issues of subjectivity and weariness. Using the bias between model simulations and measured time series could be a starting point for identifying potential anomalies. Unfortunately, to our knowledge, these techniques still require improvements. Such an algorithm should be flexible regarding the types of anomaly to identify, and might be trained for each study to avoid the risk of removing data of interest (e.g., using a visual inspection such as the one reported in this study). Ideally, hydrologists should share a common library of anomaly types such as suggested by Wilby et al. (2017). "*

I 100% agree with this statement and e.g. the automatic quality control functions in the GSIM paper yield very suboptimal results. I therefore value that the authors opted for publishing the annotations from their study, which could be beneficial when testing future approaches for automatic detection algorithms. In this regard though, I agree with the review comment by Martin Gauch, that it would be very helpful if you would also include the streamflow time series in the published data, and maybe the GR5J simulations. As Martin Gauch mentioned, the linked homepage is in French, which e.g. I do not speak. I tried to use a translation tool but after 10 minutes of trying to figure out how to download data, I gave up. It also seems like there is no API access for downloading the data, which makes the effort to get the ~600 station time series quite cumbersome. If this data already exists in the hands of the authors and there are no constraints from the data provider that prohibits the publication of their data, then why not include the streamflow time series as well. Otherwise I see limited use in the published annotations, which would be a shame.

Thanks you very much for your kind words about the paper.

As suggested to Martin Gauch's commentary (RC1):

Unfortunately, we are not allowed to provide for download data that we did not produce and that we do not own.

Regarding hydrological data, we would like to mention the existence of the hub'eau API to collect streamflow time series over France from the HydroPortail. It is a French service that aim at simplifying access to water data:

https://hubeau.eaufrance.fr/page/api-hydrometrie

A tutorial is available here (unfortunately also in French, but translation tools seem to be efficient in providing an English version):

https://hubeau.eaufrance.fr/page/api-qualite-cours-deau-tuto

We will mention the hub'eau API in the data availability section.

---

## Author Comment (AC3)

We thank Dr Gelfan for his comments about the paper.

Reviewers' comments are shown in black. Authors' responses are shown in green

During the review process, we identified an error in the computation of one of the hydrological indicators ($VCN30_5$). We corrected the error. The change rate values of the VCN30 are now more consistent with those of the QMNA (both low-flow indicators) than before. We will update the figure 7 and the values of changes rate of the $VCN30_5$ in the results section of the manuscript. We will also update the appendix C.

The study is a new attempt to reveal non-natural records of different origins, including erroneous ones, in streamflow time-series. The authors developed a comprehensive protocol for visual inspection of river flow data and involved 43 experts to detect anomalies in 674 streamflow time series in France using the protocol. The study showed a huge variability in the assessments of experts and confirmed the prevailing a priori ideas about the predominance of subjective factors when deciding on the presence of anomalies. Nevertheless, even with such uncertain results, the authors were able to formulate several recommendations, among which two seem to me to be the most important: (1) analyze as few types of anomalies as possible; and (2) allow experts to supplement the detected anomalies with confidence estimates.

Overall, I believe that the manuscript addresses relevant scientific issues and contains results that could make a useful contribution to future studies. The scientific methods and assumptions are valid and clearly outlined. The presentation is well structured and clear. I find the study to be interesting and recommend the manuscript for publication after minor revisions.

Thank you for you encouraging words about the manuscript.

Compared to Martin Gauch's excellent review already published, there is very little I could add. I fully agree with the major comments 2, 3, and 5 of this review; namely, following these comments, I also recommend the authors: to compare the obtained "change rates" with the values that would have been obtained by randomly deleting the same amount of data from the analyzed series; to evaluate the inter-evaluator agreement within certain categories of experts; and to assess whether the quality of hydrological simulations will change when evaluating the performance criterion on the cleaned series.

As many of the comments match those of Martin Gauch, we refer to our comments about the impact of random sampling of anomalies on hydrological change rates. (Answers to RC1 Martin Gauch's comments 2, 3, and 5)

Regarding the inter-evaluator, we feel that your comment goes a little further and aim at assessing if evaluators agree more with other evaluators of the same level of expertise (as times series were analyzed by two evaluators with potentially different levels of expertise). The figure below illustrates the mean (+/- standard deviation) inter-evaluator agreement for each combination of level of expertise. There is no evidence for a better combination of level of expertise that maximize the inter-evaluator agreement, even for the combination of 2 senior hydrologists (figure below).

During the experiment, we avoided the combination of 2 novice evaluators for a station, this is why "novice vs. novice" is missing from the graph.

[Figure]

In addition to the technical comments below, I would like to make two more general notes, and I'll be grateful if the authors comment on these issues in their response.

The first one concerns to the organization of the related studies. It seems logical to me to make one preparation. Before the main study begins, ask experts to weigh in on one or a few (but not many) reference streamflow time-series where some of the data has been substituted with fictitious data that the organizers are aware of. This stage will provide a preliminary general sense of the potential levels of expert agreement and the accuracy of their expert judgments.

We totally agree, the fictitious data you suggest to add to time series could be part of the inter-calibration of the evaluators phase that we suggested in the manuscript. Since our initial aim was to clean a large dataset of streamflow time series, the study of the subjectivity of the individuals and of the distribution of the anomalies came afterward. We can mention this as a recommendation in the discussion section (L344).

*"A phase of inter-calibration of evaluators, and even better with the data producer when possible, is highly recommended as it could reduce the subjectivity of such an exercise. This calibration phase could be completed by assessing the ability of the evaluator to detect fictitious anomalies in streamflow time series."*

The second general comment relates to my personal view on the perspective of visual detection of anomalies in the streamflow time-series. Given the inevitable high level of subjectivity in expert judgments (associated, first of all, with the experts' experience), I believe that expert assessments

would become more effective if not the entire series of observations were subjected to visual analysis but only its suspicious parts, previously identified using popular quantitative algorithms (k-nearest neighbors, clustering based algorithms, machine learning algorithms, etc.). This will make it possible to reduce subjectivity and increase the information content of expert analysis.

> An algorithm that identify suspicious periods seems a more achievable goal than to precisely identify time steps with anomalies, though the risk of removing data of interest remain relevant. We propose to mention that in the manuscript (L344).

> *"An automatic detection of anomalies could avoid these issues of subjectivity and weariness. As a first step, an automatic detection could identify suspicious parts of streamflow time series that would afterwards be the subject of a visual inspection, instead of inspecting the whole time series."*

**Technical comments**

Line 90: "available length of the time series greater than 25 years..." as it follows from line 96

> We will rephrase for more consistency, thanks.

> *"(3) available length of the time series greater than 26 years at a daily time step between 1976 and 2019"*

Line 138: It is unclear to me what the reason was to limit an evaluation time. It seems to me that it is more important to get a thoughtful assessment than a quick response.

> Evaluators were free to take all the time needed to inspect the time series. We provided this duration for information. We will clarify that in the manuscript to avoid any confusion.

> *"We estimated the time needed to evaluate one station to be approximately 10--15 min per evaluator, although we haven't set a time limit."*

Line 167: "...are the duration of anomaly considering the intersection and the union..., respectively."

> Nice catch! We will correct this sentence, thanks.

Fig. 3b: It is not entirely clear how the inter-evaluator agreement between an expert who analyzed data from 111 stations and another expert who processed data from a much smaller number of stations (say, 10) was established. Please clarify

> A station was always analyzed by 2 evaluators, therefor it was always possible to compute their agreement rates. Since, each evaluator analyzed from 5 to 111 stations, resulting in 5-111 agreement rates (one by station analyzed) we can draw their distribution (as displayed by the boxplots in figure 3b).

I suggest including the main recommendations formulated in subsection 5.3 and related to visual inspection of streamflow time series into the conclusions.

> We will write a short paragraph about the lessons learned from the visual inspection in the conclusion section.

*"This study also provided recommendations for future campaigns of visual inspection of time series. We strongly suggest setting up a phase of inter-calibration of evaluators in order to assess their subjectivity, as well as adding a confidence rate to the reported anomalies in order to identify more doubtful periods. Ideally, the development of automatic detection of anomalies, or at least doubtful periods, could greatly improve data cleaning stage."*

---

## Referee Report (RR1)

**Review of "On the visual detection of non-natural records in streamflow time series: challenges and impacts"**

*Iteration 2*

The authors have thoroughly answered and addressed my comments from the previous iteration. I am satisfied with the changes made in the revised version.

Martin Gauch